# Is *N*-Carbamoyl Putrescine, the Decarboxylation Derivative of Citrulline, a Regulator of Muscle Protein Metabolism in Rats?

**DOI:** 10.3390/nu11112637

**Published:** 2019-11-03

**Authors:** Prasanthi Jegatheesan, David Ramani, Mickael Lhuillier, Naouel El-Hafaia, Radji Ramassamy, Mohamed Aboubacar, Samir Nakib, Huixiong Chen, Christiane Garbay, Nathalie Neveux, Cécile Loï, Luc Cynober, Jean-Pascal de Bandt

**Affiliations:** 1EA4466, Faculty of Pharmacy, Paris Descartes University, 75270 Paris, France; pira_jegatheesan@hotmail.com (P.J.); d.ramani@ch-mantes-la-jolie.fr (D.R.); mickael.lhuillier@parisdescartes.fr (M.L.); naouel.el-hafaia@parisdescartes.fr (N.E.-H.); radji.ramassamy@parisdescartes.fr (R.R.); mohamed.aboubacar@parisdescartes.fr (M.A.); samir.nakib@aphp.fr (S.N.); nathalie.neveux@aphp.fr (N.N.); cecile.loi@citrage.fr (C.L.); luc.cynober@parisdescartes.fr (L.C.); 2Clinical Chemistry Department, Hôpital Cochin, AP-HP, 75679 Paris, France; 3Laboratory of Pharmacologic and Toxicologic Chemistry and Biochemistry, UMR 8601 CNRS, Paris Descartes University, 75270 Paris, France; huixiong.chen@parisdescartes.fr (H.C.); christiane.garbay@parisdescartes.fr (C.G.)

**Keywords:** *N*-carbamoyl putrescine, citrulline, muscle anabolism, catabolism, malnutrition

## Abstract

*N*-carbamoyl putrescine (NCP), the decarboxylation derivative of citrulline, metabolically related to polyamines, may exert biological effects in mammals. The aim of this study was (i) to evaluate the nutritional properties of NCP in healthy rats and (ii) to determine the effect of NCP administration on muscle metabolism in malnourished old rats. The nutritional properties of NCP were first evaluated in 20 8-week-old male rats randomized to receive for two weeks a standard diet either alone (C group) or supplemented with NCP, 5 or 50 mg/kg/d. In a second study, 29 malnourished 18-month-old male rats were studied either before or after a 4-day refeeding with a standard diet either alone (REN group) or supplemented with NCP, 1 or 10 mg/kg/d. NCP had no effect on weight gain and body composition in either of the two studies. In healthy rats, muscle protein content was significantly increased in the soleus with NCP 5 mg/kg/d. A decrease in plasma glutamine and kidney spermine was observed at the 50 mg/kg/d dose; otherwise, no significant changes in plasma chemistry and tissue polyamines were observed. In malnutrition-induced sarcopenic old rats, refeeding with NCP 10 mg/kg/d was associated with higher tibialis weight and a trend for increased protein content in extensor digitorum longus (EDL). While the muscle protein synthesis rate was similar between groups, ribosomal protein S6 kinase was increased in tibialis and higher in the EDL in NCP-treated rats. The muscle RING-finger protein-1 expression was decreased in tibialis and urinary 3-methyl-histidine to creatinine ratio slightly lower with the supply of NCP. However, this initial period of refeeding was also associated with elevated fasted plasma triglycerides and glucose, significant in NCP groups, suggesting glucose intolerance and possibly insulin resistance. NCP was well-tolerated in healthy young-adults and in malnourished old rats. In healthy adults, NCP at 5 mg/kg/d induced a significant increase in protein content in the soleus, a type I fiber-rich muscle. In malnourished old rats, NCP supply during refeeding, may help to preserve lean mass by limiting protein breakdown; however, these effects may be limited in our model by a possible immediate refeeding-associated glucose intolerance.

## 1. Introduction

Citrulline has been shown to promote the preservation of lean body mass in aging animals, probably in part through its role as a precursor of arginine which has been shown to display beneficial effects, notably on endothelial functions, immune status and glucose and lipid metabolism [1].

While arginine and citrulline properties are not fully similar, they are both precursors of polyamines (Figure 1). Aliphatic polyamines (putrescine, spermidine and spermine) are polycationic molecules derived from ornithine decarboxylation. Through their role in cellular proliferation, they are involved in multiple physiological functions, such as immunity, wound healing and gut trophicity [2]. Another polyamine, agmatine, derived from arginine decarboxylation, has been shown to activate endothelial nitric oxide (NO) synthase, to stimulate insulin and catecholamine secretion. Putrescine and agmatine are decarboxylation derivatives of ornithine and arginine respectively, two key amino acids (AAs) of the urea cycle. Citrulline is also an AA belonging to this cycle, and presents a decarboxylation derivative, *N*-carbamoyl putrescine (NCP), a metabolic intermediate in putrescine synthesis in plants [3] and bacteria [4,5,6]. However, NCP metabolism has not been characterized in mammals and its biological effects and toxicity have not been evaluated.

In this study, we hypothesized that, as with the amines derived from ornithine and arginine that participate in the metabolic properties of these amino acids, NCP could participate in the anabolic effects of citrulline. We first evaluated the nutritional properties of NCP in healthy young-adult rats, its tolerance and its effects on polyamine metabolism. Thereafter, given the positive effect of NCP on muscle protein content, we evaluated its effect on muscle protein metabolism in a model of malnutrition-induced sarcopenia in old rats. We also investigated the signaling pathways involved in the balance between muscle anabolism and catabolism. Doses of NCP administered were initially based on the mean dietary intakes of aliphatic polyamines, as described in the literature [7,8,9]: a 5 mg/kg/d dose corresponds to the daily intake of putrescine in rat, the molecule metabolically closest to NCP, and a 50 mg/kg/d dose to a presumably pharmacological dose. The doses used in the second study were adapted according to the results of the first study.

## 2. Materials and Methods

### 2.1. Animals

Twenty young-adult (eight-week old) and twenty-nine old (eighteen-month old) male Sprague–Dawley rats (Charles Rivers, Lyon, France) were used. They were acclimatized for two weeks to our animal facility with free access to standard rodent chow (841202-M20, Dietex, France: 18% protein, 3% fat, 57% carbohydrate, 4% fiber, 12% water, vitamins and minerals; 1234 kJ/100 g) either as pellets or in powder form and with water ad libitum. Food consumption was measured during the acclimatization period in order to provide exactly 100% of daily intakes during the experimentation periods.

Animal care and experimentation complied with French and European Community regulations for animal care and experimentation. The study protocol has been approved by the Regional ethic committee of Ile-de-France (CE2A-34, project number 16-084).

NCP was synthesized and purified as described previously [10].

### 2.2. Experimental Design

Study 1: After acclimatization, young-adult rats were randomized into three groups to receive for two weeks a standard diet either alone (*n* = 6; control group) or supplemented with NCP, either 5 mg/kg/d (*n* = 7, NCP5 group) or 50 mg/kg/d (*n* = 7; NCP50 group). Diet was administered at 100% of daily intakes measured during the acclimatization period, and NCP, when given, was mixed with the powdered rodent chow.

Study 2: Malnutrition was induced in old rats by a dietary restriction to 50% of their spontaneous intakes (a model previously validated in our laboratory [11]) for 6 weeks. The animals were randomized into four groups: at the end of the malnutrition period, 8 rats (*n* = 8, DEN group) were sacrificed immediately, while the animals of the other groups were housed individually in metabolic cages and received, for 4 days, their standard diet at 100% of their spontaneous food intake alone (*n* = 9, REN group) or combined with NCP at 1 mg/kg/d (*n* = 9, NCP1 group) or 10 mg/kg/d (*n* = 8, NCP10 group). For a precise administration of NCP, it was given as an aqueous solution in 5 mL water supplied every morning in parallel with the daily food ration and the water supply was withdrawn until full consumption was achieved.

Animal weight, behavior and mortality were monitored throughout the experimental period. Complete consumption of the daily food ration was also checked. At the end of the experimental period, both young and old rats, in the fasted state, were anesthetized by isoflurane inhalation and euthanized by decapitation.

A measurement of protein synthesis was performed in all malnourished old rats just prior to euthanasia. In brief, animals were anesthetized as described above and then given an intraperitoneal injection of 22 mg/kg puromycin. At exactly 30 min after injection, rats were euthanized, and blood and tissue samples were taken.

### 2.3. Sample Treatment

Mixed blood was collected onto heparinized tubes and rapidly centrifuged (5,000 rpm, 10 min at 4 °C). For plasma AA determination, samples were deproteinized with sulfosalicylic acid (30 mg/mL).

Only in young-adult rats, the jejunum and ileum were taken and were washed with cold NaCl (0.9%) and reverted. Thereafter, the intestinal mucosa was scraped, rapidly frozen in liquid nitrogen and stored at −80 °C until analysis.

In all rats, the liver was immediately removed and weighed, and a sample was cut off, frozen in liquid nitrogen and stored at −80 °C until analysis.

Two (study 1) or three (study 2) muscles of the hindlimb, the tibialis (a mix muscle with both type I and II fibers), the soleus (rich in type I fibers) and the extensor digitorum longus (EDL, rich in type II fibers) were rapidly removed, weighed, frozen in liquid nitrogen and stored at −80 °C until analysis.

Tissues samples (0.1 g/mL) were homogenized in 10% trichloracetic acid and 0.5 mM EDTA using an Ultraturrax T25 (IKA Labortechnik, Stauffen, Germany). After centrifugation, supernatants were collected for AA, NCP and polyamine determination, and stored at −30 °C until analysis. After delipidation with methanol/ether (1:1 *v*/*v*), pellets were resuspended in 1 N NaOH (3.3 mL for 100 mg tissue) and incubated overnight at 40 °C. Protein content was assayed using Gornall’s method and a Genesys spectrophotometer (ThermoSpectronic, New York, NY, USA).

### 2.4. Nitrogen Metabolism

Nitrogen homeostasis was estimated by the determination of nitrogen balance [12]. Nitrogen was quantified by chemiluminescence using an Antek 9000 apparatus (Alytech, Houston, Texas, USA) and the nitrogen balance was calculated as the difference between nitrogen intake and nitrogen urinary output, assuming similar cutaneous and digestive nitrogen losses in the different groups of rats.

For the evaluation of protein catabolism, urinary 3-methylhistidine (3MH) and creatinine excretion were measured as previously described [13]. Myofibrillar protein degradation was evaluated by urinary 3MH to creatinine ratio [13].

For the quantification of protein synthesis, relative protein synthesis rate was evaluated in muscles by the SunSET method and quantified by western blot (see general procedure below) using anti-puromycin antibodies (1:1000, clone 12D10; Millipore).

### 2.5. Expression/Activation of Anabolic and Catabolic Signaling Protein 

Expression and activation statuses of anabolic and catabolic signaling proteins were determined according to the technique previously described [14]. Muscles samples were ground in liquid nitrogen to obtain a powder that was homogenized in an ice-cold buffer (HEPES pH 7,5, NaCl, EDTA, β-glycerophosphate, NaF, orthovanadate, triton X-100, protease and phosphatase inhibitor cocktail). After centrifugation (30 min, 13,000× *g*, 4 °C), the supernatants were collected. Protein levels were determined using the Bicinchoninic acid assay (Protein Assay kit; Interchim). Proteins were then analyzed by western blot. The primary antibodies used were from Cell Signaling: rabbit anti-phospho4E-BP1 (Ser65), (diluted 1:1000), rabbit anti-4E-BP1 (1:1000), phosphoS6K1 (Thr389) (1:1000), rabbit anti-S6K1 (1:1000), rabbit anti-Murf1 (1:500) and rabbit anti-GAPDH (1:2000). After three washes with TBS-1% Tween 20 buffer, protein staining was done with incubation with peroxidase-conjugated goat anti-rabbit IgG (1:5000; Dakocytomation) for 1 h at room temperature and the addition of a chemiluminescence reagent (Amersham ECL Prime Western Blotting Detection Reagent, GE Healthcare). Protein bands were quantified by densitometry using ImageJ software. Protein expression levels were normalized to that of GAPDH. For S6K1 and 4E-BP1, protein activation was calculated as the ratio of the phosphorylated protein to its total expression.

### 2.6. Biological Assessment

Plasma total proteins, urea, creatinine, uric acid, glucose, cholesterol, triglycerides (TG) and bilirubin, and activities of aminotransferases (aspartate aminotransferase (AST), alanine aminotransferase (ALT)), alkaline phosphatase (ALP), creatine kinase (CK) and lactate dehydrogenase (LD)), were assayed on a Roche Diagnostic multiparametric analyzer using dedicated reagents (Roche Diagnostic, Meylan, France). AAs were measured as previously described [13]. Creatinine clearance was calculated as urinary creatinine × urinary flow/plasma creatinine.

For NCP and polyamine determination, an adaptation of the HPLC method developed by Seiler [15] was used [10].

### 2.7. Calculations and Statistical Analysis

Sum of branched-chain AAs (ΣBCAA) was calculated as the sum of leucine, valine and isoleucine. The sum of non-essential AAs (ΣNEAA) was calculated as the sum of alanine, arginine, asparagine, aspartate, cystine, glutamine, glutamate, glycine, proline, serine and tyrosine. The sum of essential amino acids, ΣEAA, was calculated as the sum of threonine, methionine, valine, leucine, isoleucine, phenylalanine, histidine and lysine. Total plasma amino acid level, ΣAA, was calculated as the sum of ΣNEAA and ΣEAA.

Data are expressed as means ± SEMs. Statistical analyses were performed with Prism 6.0 (GraphPad® software, San Diego, CA, USA). Homogeneity of variance of data was assessed using Bartlett’s test. Differences between groups were analyzed using one-way ANOVA followed by Fisher’s PLSD test (1st Study) or Tukey–Kramer (2nd study). In the case of heterogeneity of variance, a Kruskal–Wallis test followed by Dunn’s test was performed. For all the tests, *p* < 0.05 was considered significant.

## 3. Results

Regardless of the study, no rat showed abnormal behavior, and there was no mortality.

### 3.1. Study 1

#### 3.1.1. Tolerance

All young-adult animals achieved normal weight gain and there was no difference between rats receiving NCP and controls (Table 1). We did not observe any difference in plasma glucose, creatinine, TG or cholesterol, nor in enzyme activities (Table 1 and data not shown). NCP at the 50 mg/kg/d dose significantly increased plasma urea compared to 5 mg/kg/d (controls: 3.8 ± 0.4 mmol/l, NCP5: 3.4 ± 0.3 mmol/l and NCP50: 4.1 ± 0.6 mmol/l; *p* = 0.015 NCP50 versus NCP5).

#### 3.1.2. Nitrogen and Muscle Protein Metabolism

Muscle weight did not differ among groups, but muscle protein content was significantly increased in the soleus in NCP5 group compared to control group (Table 2).

Analysis of the plasma AAs showed that plasma glutamine level was significantly lower in the NCP50 group than in controls (control: 716 ± 139 µmol/l, NCP5: 652 ± 96 µmol/l and NCP50: 591 ± 65 µmol/l; *p* = 0.043 NCP50 versus control) and there was a trend for a NCP dose–effect relationship on GLN levels (*p* = 0.06). Plasma ornithine showed a similar dose–effect relationship (control: 51 ± 11 µmol/l, NCP5: 47 ± 4 µmol/l and NCP50: 41 ± 9 µmol/l; *p* = 0.055). No differences were observed for muscle ΣBCAA, ΣEAA, ΣNEAA or ΣAA (data not shown).

#### 3.1.3. Polyamine Metabolism

Plasma polyamines, putrescine, spermidine and spermine, were similar between the three groups (Table 3). In the kidney, a dose dependent (*p* = 0.01) decrease in spermine was observed but no difference in putrescine and spermidine. In muscles, putrescine tended to be higher in the soleus in NCP50 group than in NCP5 group (*p* = 0.10), but the tibialis was not affected. Spermine levels tended to be lower in tibialis in NCP5 group versus controls (*p* = 0.07), but were unaffected in soleus; muscle spermidine was similar between the three groups in soleus and tibialis. Last, no difference in polyamine levels was observed between the three groups in the liver, nor in jejunal or ileal mucosa. No NCP was detected in the plasma and in any of the tissues tested (data not shown).

### 3.2. Study 2

#### 3.2.1. Tolerance

Refeeding-induced weight gain was not influenced by NCP (Table 4).

Plasma glucose tended to increase with refeeding, and this reached statistical significance in NCP-supplemented groups. Refeeding was associated with a significant increase in plasma TG with or without NCP administration (Table 4).

Liver and renal function were similar between the 4 groups (Table 4).

#### 3.2.2. Nitrogen and Muscle Protein Metabolism

Refeeding was associated with an increase in nitrogen balance and this was not significantly influenced by NCP. Whatever the dose, NCP increased the weight of tibialis (*p* = 0.058) but not of EDL and soleus. Moreover, NCP tended to increase the protein content of the EDL (ANOVA *p* = 0.067; NCP10 versus DEN, *p* = 0.05) without effect on the other muscles (Table 5).

No significance difference was noted in relative muscle protein synthesis rate evaluated by puromycin incorporation (ANOVA *EDL*: *p* = 0.25; soleus: *p* = 0.09; tibialis: *p* = 0.91) (data not shown). While S6K protein activation was not modified following NCP supplementation, no matter the muscle (ANOVA *p* = 0.09), 4E-BP1 phosphorylation was significantly increased by refeeding in EDL (Figure 2).

Refeeding was associated with an increase in urinary 3MH/creatinine ratio; it was slightly lower in NCP-treated groups compared to REN group, but this did not reach statistical significance (Table 5). While variations of Murf1 expression were not significant in EDL, it was increased by refeeding in soleus and tibialis and this was prevented in NCP1 and NCP10 groups (Figure 3).

Plasma ΣBCAA, ΣEAA, ΣNEAA and ΣAA showed only limited differences between the four groups with a decrease in cystine in the REN and NCP10 groups and glycine in the three refed groups (Table 5 and data not shown). Only limited differences in muscle AA contents were observed (Table 6).

Refeeding was associated with significantly increased liver weight without significant difference in protein content (Table 5).

## 4. Discussion

In this study aimed at assessing the nutritional properties of NCP, we showed that this citrulline derivative allows a muscle protein gain in young animals. Unlike amines derived from other amino acids of the urea cycle, NCP appears to affect the metabolism of aliphatic polyamines only very moderately. Finally, our experiments showed encouraging effects in terms of reduction of catabolism in a model of sarcopenia induced by prolonged undernutrition in old rats. However, in this model, the effects of NCP may be partially limited by the important metabolic disturbances associated with the early stages of refeeding.

First, it is important to note that, globally, our data indicate that NCP administration in rats, at the levels tested, was not associated with toxicity both in the short term in old rats or after a two-week administration in young-adult rats. Indeed, we observed neither mortality nor alterations in liver, renal or muscle functions, or in overall metabolism. It can be compared with the toxicity of putrescine, NCP’s closest derivative: Til et al. [9] have shown that the approximate LD_50_ of putrescine is 2000 mg/kg, and the non-observed-adverse effect level of putrescine, when given as a supplement in the diet for six weeks, is 180 mg/kg/day. Those levels are far higher than the NCP levels tested in our study, even when taking into account the fact that NCP may be metabolized into putrescine or its metabolites by the microbiota in the gut lumen. Acute and subacute toxicity studies using higher NCP levels may help to determine the LD_50_ and non-observed-adverse effect levels of NCP. However, NCP50 was associated with a slight increase in plasma urea and a decrease in plasma glutamine. This may suggest increased glutamine degradation for urea synthesis. Of note, it has been shown [16] that spermine activates hepatic glutaminase, leading to an increase in glutamine uptake by the liver and an increase flux of ammonia in the liver, activating in turn carbamyl-phosphate synthase [16], the first enzyme of the urea cycle. However, in our experiment the increase in liver spermine did not reach statistical significance. As glutamine availability is an important factor for immunity and protein metabolism, doses of NCP below 50 mg/kg were chosen for our second study.

In young-adult rats, on a nutritional point of view, NCP administration was not associated with modifications in body weight gain compared to the controls. Interestingly, the soleus’s protein content was significantly higher in the NCP5 group than in the control group. The absence of effect on the tibialis muscle is probably related to muscle fiber composition. Indeed, compared to fast twitch muscles, slow twitch muscles tend to have a low activity level of ATPase and a slower contraction speed with a less-developed glycolytic capacity. They have been shown to contain high concentrations of mitochondrial enzymes; they are, thus, more resistant to fatigue. Importantly, it has been shown that protein synthesis is differently regulated depending on the muscle fiber type [17]. For example, muscle enrichment in type I fibers is favored by endurance exercise and thwarted by physical inactivity. Muscle enrichment in type I fibers is considered a favorable factor in the prevention of metabolic syndrome and related disorders.

Last, unlike agmatine, which inhibits cellular uptake and the synthesis of polyamines [18,19] and induces their use by spermidine/spermine-acetyltransferase [19,20], NCP has only limited effects on liver, plasma and intestine polyamine contents in young-adult rats. The higher dose of NCP (50 mg/kg/d) decreased spermine content in rat kidney, without changes in spermidine and putrescine contents. NCP’s effects on polyamine content in muscle were less marked and did not show a dose-dependent pattern. Tissue-specific changes in polyamine content, varying with the polyamine tested, have already been shown by other authors [21,22]. A possible explanation for a selective effect of NCP on kidney spermine may be that the kidney, as a potential main organ for NCP excretion, has been exposed to higher local concentrations of NCP than other organs, which may suggest an effect of NCP concentrations at high doses. Moreover, as only spermine is specifically affected, NCP may possibly act on the enzymes of polyamine interconversion. A pharmacokinetic study of NCP and studies involving NCP labeled with a radioactive isotope are needed to identify the different metabolic pathways involved in this metabolism.

In malnourished old rats, NCP was well-tolerated without any effect on plasma glutamine, ornithine or urea, unlike the dose of 50 mg/kg/d, suggesting that the dosage levels were the right ones. Unlike muscle protein gain in young-adult rats, the results in malnourished old rats were relatively disappointing. This is probably due in part to a lack of power in our study; unfortunately, even without taking into account the high cost of old rats, their availability for experimentation is very limited, making this type of work difficult. Many results suggested a potentially positive effect of NCP at the 10 mg/kg/d dose on protein metabolism, but the differences rarely reached statistical significance. Thus, overall, muscle protein content seemed significantly higher, but this increase was only at the limit of significance in the EDL. Similar results can be found for the activating phosphorylation of S6K, an important regulator of protein synthesis, which tended to increase in EDL. In addition, the activation of proteolysis by refeeding appeared to be 20% lower with NCP 10 mg/kg/d than in REN group, but again, without reaching statistical significance. In parallel, the activation of Murf1, an important regulator of proteolysis, was significantly lower with NCP in two of the three muscles studied. It should be noted that the effect of NCP seems more marked on slow-twitch muscle in study 1, and on fast-twitch muscle in study 2. Several factors may be involved besides the fact that none of the muscles studied exclusively contain a single type of fiber. Indeed, fast-twitch muscles (EDL, tibialis) are more influenced by dietary restrictions than slow-twitch muscles (soleus). Moreover, the respective percentages of the different types of fiber are affected by aging, and at the same time, older rats have a lack of adaptation to long-term dietary restrictions. It would, therefore, be interesting to study which type of fiber is specifically affected.

Several elements can probably explain the limited effect of NCP in these experiments. First, the muscle protein gain observed in young rats was 18 mg/g in about two weeks. We decided to conduct this study in the elderly rat over 4 days due to the fact that a significant result on this endpoint was observed with citrulline. Unfortunately, in our current study, we failed to obtain a significant result to that endpoint with NCP. Regarding muscle protein synthesis and transduction mechanisms, these parameters were evaluated in the fasting state. The choice of fasting animals was a consequence of the difficulty of handling these malnourished old animals and the impossibility of administering to them a test meal by gavage to study the postprandial situation. Finally, refeeding, especially in the presence of NCP, is associated with glucose intolerance. Although plasma insulin levels have not been determined and a decrease in insulin secretion cannot be ruled out; the very significant increase in blood glucose and plasma triglycerides while animals were in a strict fasting state is in favor of insulin resistance. The first few days after the return to normal nutrition in malnourished animals are indeed associated with metabolic changes constituting the so-called “catch-up fat” phenomenon corresponding to a preferential restoration of lipid reserves. During this period, animals show a decrease in thermogenesis and muscle glucose use and a reduction in muscle protein synthesis [23,24]. This phenomenon is probably all the more marked given that the animals were refed at 100% of their spontaneous food intake measured before semi-starvation. This could also have affected the effectiveness of NCP.

Additional experiments over a longer period of time of refeeding, with a more controlled dietary intake, and evaluating the post-prandial situation, are needed to confirm the potentially beneficial effect of NCP on protein metabolism.

## 5. Conclusions

NCP, at the doses studied, is well-tolerated in healthy young-adult and malnourished old rats. In young-adult animals, NCP 5 mg/kg/d administration for two weeks was associated with a significant increase in protein content in soleus, a type I fiber-rich muscle. In malnourished old rats, our data suggest that NCP may act at least by limiting protein breakdown; however, these effects may be dampened in this model of semi-starvation–refeeding by the immediate refeeding effect and a possible catch-up fat-induced insulin resistance. Additional studies are needed to confirm NCP’s effectiveness on muscle protein metabolism in situations of malnutrition and to investigate the mechanism involved.

## Figures and Tables

**Figure 1 nutrients-11-02637-f001:**
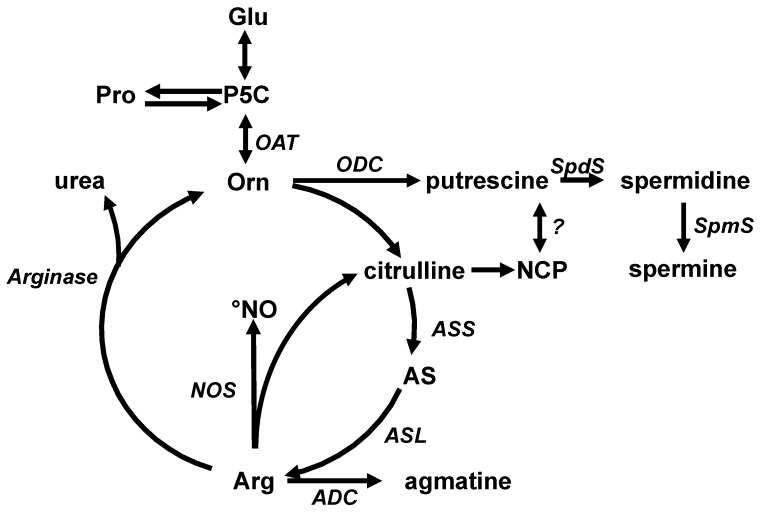
Citrulline, arginine and ornithine as precursors of potentially active amine derivatives. ADC: Arg decarboxylase; AS: argininosuccinate; AS: AS synthase; ASL: AS lyase; OAT: Orn aminotransferase; OCT: Orn carbamyl transferase; ODC: Orn decarboxylase; NOS: nitric oxide synthase; SpdS: spermidine synthase; SpmS: spermine synthase; ?: demonstrated in some vegetal cells and bacteria.

**Figure 2 nutrients-11-02637-f002:**
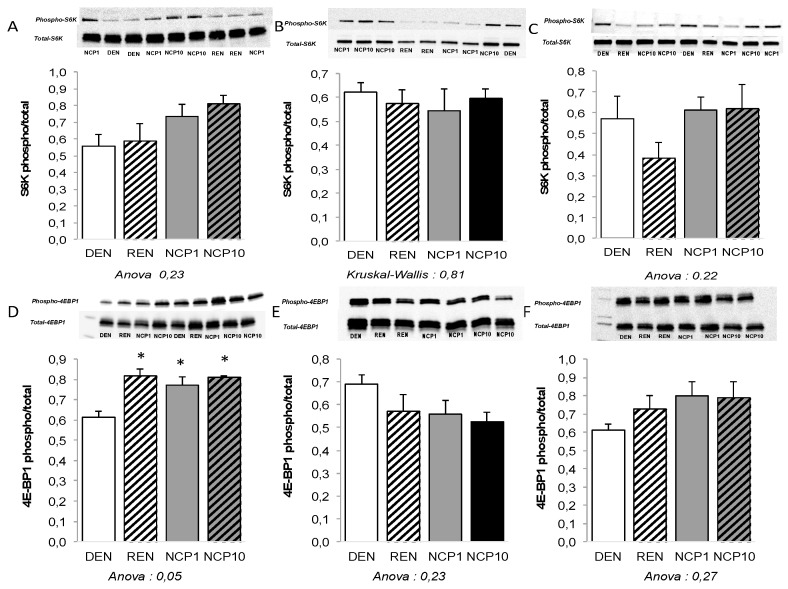
The effect of NCP on the mammalian muscle target of rapamycine complex 1 (mTORC1) signaling during refeeding of malnourished old rats. Malnutrition was induced in old rats by a dietary restriction to 50% of their spontaneous intakes for 6 weeks. Thereafter, eight rats (*n* = 8, DEN group) were sacrificed immediately, while the animals of the other groups received, for 4 days, their standard diet at 100% of their spontaneous food intake alone (*n* = 9, REN group) or with NCP, 1 mg/kg/d (*n* = 9, NCP1 group) or 10 mg/kg/d (*n* = 8, NCP10 group). Phosphorylation status of p70 ribosomal protein S6 kinase 1 (S6K) (**A**,**B**,**C**) and eukaryotic initiation factor 4E-binding protein 1 (4E-BP1) (**D**,**E**,**F**) were measured at the end of this 4-day period in fasted animals by western blot in the extensor digitaris longus (**A**,**D**), soleus (**B**,**E**) and tibialis (**C**,**F**). All values (means ± SEMs) are expressed as the ratios of each phosphorylated protein to their total expression. One-way ANOVA and post-hoc Tukey–Kramer; statistical significance: *p* < 0.05. * *p* < 0.05 versus DEN.

**Figure 3 nutrients-11-02637-f003:**
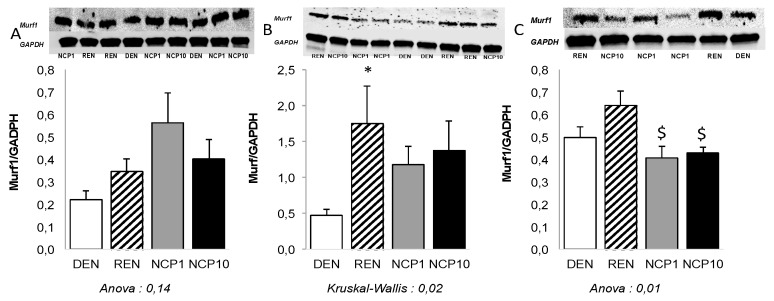
The effect of NCP on expression of Murf1 during refeeding of malnourished old rats. Malnutrition was induced in old rats by a dietary restriction to 50% of their spontaneous intakes for 6 weeks. Thereafter, eight rats (*n* = 8, DEN group) were sacrificed immediately while the animals of the other groups received, for 4 days, their standard diet at 100% of their spontaneous food intake alone (*n* = 9, REN group) or with NCP, 1 mg/kg/d (*n* = 9, NCP1 group) or 10 mg/kg/d (*n* = 8, NCP10 group). Expression of Murf1 was measured at the end of this 4-day period, in fasted animals, in their extensor digitaris longus (**A**), soleus (**B**) and tibialis (**C**) by western blot. All values (means ± SEMs) are expressed as ratios to a reference gene (*GAPDH*). One-way ANOVA and post-hoc Tukey–Kramer or Kruskal–Wallis and post-hoc Dunn test; statistical significance: *p* < 0.05. * *p* < 0.05 versus DEN group. $ *p* < 0.05 versus REN group.

**Table 1 nutrients-11-02637-t001:** NCP tolerance in adult rats.

	C	NCP5	NCP50	*p* Values
Weight (g)	314 ± 4	313 ± 7	316 ± 5	0.93
Weight gain (g)	51 ± 3	48 ± 3	51 ± 3	0.76
**Metabolism**				
Glucose (mmol/l)	6.6 ± 0.3	7.1 ± 0.3	7.0 ± 0.3	0.37
Cholesterol (mmol/l)	1.4 ± 0.1	1.3 ± 0.1	1.3 ± 0.1	0.79
Triglycerides (mmol/l)	0.5 ± 0.1	0.5 ± 0.0	0.6 ± 0.1	0.22
**Liver, muscle and kidney function**				
AST (IU/L)	152 ± 6	163 ± 14	162 ± 13	0.78
ALT (IU/L)	46 ± 2	46 ± 3	45 ± 3	0.93
Creatinine (µmol/l)	13 ± 1	12 ± 1	13 ± 1	0.28
Creatine kinase (UI/L)	11 ± 1	11 ± 2	11 ± 1	0.99
Bilirubin (µmol/l)	1.0 ± 0.3	1.1 ± 0.1	1.0 ± 0.0	0.76

Young adult rats were randomized into three groups to receive, for two weeks a standard diet either alone (*n* = 6; C group) or supplemented with NCP, either 5 mg/kg/d (*n* = 7, NCP5 group) or 50 mg/kg/d (*n* = 7; NCP50 group). NCP, when given, was mixed with the powdered rodent chow. All the parameters, excepted weight gain, were measured at the end of the two-week feeding period in the fasted state. Weight gain was calculated for the whole feeding period. ALT: alanine aminotransferase; AST: aspartate aminotransferase. All values are presented as mean ± SEM. One-way ANOVA and post-hoc Fisher PLSD; statistical significance: *p* < 0.05.

**Table 2 nutrients-11-02637-t002:** The effect of NCP on nitrogen and protein metabolism in adult rats.

	C	NCP5	NCP50	*p* Values
**Soleus**				
Weight (g)	0.14 ± 0.00	0.15 ± 0.01	0.14 ± 0.00	0.31
Protein content (mg/g)	127 ± 7	145 ± 3 *	140 ± 3	0.05
Protein content (mg)	17 ± 1	21 ± 1 *	20 ± 0.5	0.03
**Tibialis**				
Weight (g)	0.59 ± 0.01	0.62 ± 0.02	0.60 ± 0.01	0.43
Protein content (mg/g)	174 ± 3	175 ± 3	177 ± 2	0.32
Protein content (mg)	52 ± 1	52 ± 1	53 ± 1	0.62
**Liver**				
Weight (g)	9.20 ± 0.23	9.50 ± 0.28	9.10 ± 0.13	0.49
Protein content (mg/g)	205 ± 7	211 ± 2	208 ± 5	0.90
**Plasma amino acids**				
ΣBCAA (µmol/l)	463 ± 39	422 ± 23	393 ± 32	0.32
ΣEAA (µmol/l)	916 ± 69	843 ± 30	806 ± 50	0.33
ΣNEAA (µmol/l)	2369 ± 158	2234 ± 103	2131 ± 95	0.39
ΣAA (µmol/l)	3285 ± 227	3077 ± 130	2937 ± 143	0.36

Young adult rats were randomized into three groups to receive, for two weeks a standard diet either alone (*n* = 6; C group) or supplemented with NCP, either 5 mg/kg/d (*n* = 7, NCP5 group) or 50 mg/kg/d (*n* = 7; NCP50 group). NCP, when given, was mixed with the powdered rodent chow. All the parameters (mean ± SEM) were measured at the end of the two-week feeding period in the fasted state. ΣBCAA: sum of branched-chain amino acids; ΣEAA: sum of essential amino acids; ΣNEAA: sum of nonessential amino acids; ΣAA: total plasma amino acid level. One-way ANOVA and post-hoc Fisher PLSD; statistical significance: *p* < 0.05. * *p* < 0.05 NCP5 versus C.

**Table 3 nutrients-11-02637-t003:** NCP and plasma and tissue polyamines in adult rats.

	C	NCP5	NCP50	*p* Values
**Liver (nmol/g)**				
Putrescine	5.95 ± 0.65	5.70 ± 0.83	4.79 ± 0.67	0.51
Spermidine	790 ± 32	805 ± 32	781 ± 30	0.86
Spermine	918 ± 86	930 ± 105	1103 ± 81	0.31
**Plasma (μmol/L)**				
Putrescine	0.40 ± 0.05	0.69 ± 0.16	0.56 ± 0.12	0.39
Spermidine	7.65 ± 0.68	7.83 ± 0.36	9.27 ± 0.86	0.19
Spermine	0.63 ± 0.03	0.67 ± 0.06	0.74 ± 0.04	0.23
**Jejunum (nmol/g)**				
Putrescine	84 ± 9	87 ± 10	97 ± 6	0.54
Spermidine	1884 ± 352	1551 ± 101	1677 ± 114	0.53
Spermine	1508 ± 135	1490 ± 119	1598 ± 167	0.85
**Ileum (nmol/g)**				
Putrescine	66 ± 7	79 ± 10	68 ± 7	0.51
Spermidine	1198 ± 85	1385 ± 85	1210 ± 70	0.20
Spermine	1109 ± 80	1175 ± 137	1112 ± 103	0.90
**Kidney (nmol/g)**				
Putrescine	13 ± 2	16 ± 2	20 ± 3	0.22
Spermidine	948 ± 32	954 ± 45	846 ± 69	0.28
Spermine	848 ± 85	881 ± 71	555 ± 59 *^,#^	0.01
**Soleus (nmol/g)**				
Putrescine	11 ± 2	8 ± 1	13 ± 2	0.10
Spermidine	205 ± 13	223 ± 9	210 ± 9	0.47
Spermine	164 ± 43	235 ± 35	206 ± 46	0.48
**Tibialis (nmol/g)**				
Putrescine	2 ± 0	1 ± 0	2 ± 0	0.10
Spermidine	57 ± 3	53 ± 2	54 ± 3	0.47
Spermine	430 ± 32	325 ± 29	392 ± 29	0.07

Young adult rats were randomized into three groups to receive, for two weeks a standard diet either alone (*n* = 6; C group) or supplemented with NCP, either 5 mg/kg/d (*n* = 7, NCP5 group) or 50 mg/kg/d (*n* = 7; NCP50 group). NCP, when given, was mixed with the powdered rodent chow. All the parameters were measured at the end of the two-week feeding period in the fasted state. Results (mean ± SEM) are expressed as μmol/L or as nmol/g tissue. One-way ANOVA and post-hoc Fisher PLSD; statistical significance: *p* < 0.05. * *p* < 0.05 versus C; ^#^
*p* < 0.05 versus NCP5.

**Table 4 nutrients-11-02637-t004:** NCP tolerance during refeeding of malnourished old rats.

	DEN	REN	NCP1	NCP10	*p* Values
Weight (g)	550 ± 21	604 ± 21	577 ± 19	592 ± 15	0.24
Weight gain (g)		18 ± 2 *	20 ± 3 *	13 ± 3 *	<0.0001
**Metabolism**					
Glucose (mmol/l)	11.8 ± 1.1	15.5 ± 1.6	17.1 ± 1.5 *	17.2 ± 1.1 *	0.04
Cholesterol (mmol/l)	3.6 ± 0.2	3.0 ± 0.2	3.0 ± 0.2	3.2 ± 0.2	0.11
Triglycerides (mmol/l)	0.68 ± 0.1	0.95 ± 0.09 *	1.13 ± 0.07 *	1.14 ± 0.08 *	0.002
**Liver, muscle and kidney function**					
AST (IU/L)	148 ± 18	179 ± 21	172 ± 60	196 ± 35	0.86
ALT (IU/L)	53 ± 7	98 ± 21	64 ± 16	95 ± 29	0.30
Bilirubin (µmol/l)	1.3 ± 0.2	0.9 ± 0.1	1.2 ± 0.3	1.4 ± 0.2	0.38
Creatinine (µmol/l)	29 ± 1	28 ± 2	27 ± 3	27 ± 1	0.87
Urinary Creatinine (µmol/24h)	91 ± 9	91 ± 11	74 ± 5	96 ± 7	0.31
Creatinine clearance (mL/h)	131 ± 16	140 ± 20	122 ± 14	148 ± 13	0.70

Malnutrition was induced in old rats by a dietary restriction to 50% of their spontaneous intakes for 6 weeks. Thereafter, eight rats (*n* = 8, DEN group) were sacrificed immediately while the animals of the other groups received, for 4 days, their standard diet at 100% of their spontaneous food intake alone (*n* = 9, REN group) or with NCP, 1 mg/kg/d (*n* = 9, NCP1 group) or 10 mg/kg/d (*n* = 8, NCP10 group). All the parameters (means ± SEMs), excepted weight gain, were measured at the end of the 4-day feeding period in the fasted state. Weight gain was calculated for the whole feeding period. ΣBCAA: sum of branched-chain amino acids; ΣEAA: sum of essential amino acids; ΣNEAA: sum of nonessential amino acids; ΣAA: total plasma amino acid level; ALT: alanine aminotransferase; AST: aspartate aminotransferase. One-way ANOVA and post-hoc Tukey–Kramer; statistical significance: *p* < 0.05. * *p* < 0.05 versus DEN.

**Table 5 nutrients-11-02637-t005:** The effect of NCP on nitrogen and protein metabolism during the refeeding of malnourished old rats.

	DEN	REN	NCP1	NCP10	*p* Values
**Nitrogen metabolism**					
Nitrogen balance ^a^ (mg/24h)	172 ± 32	483 ± 22 *	471 ± 27 *	418 ± 44 *	<0.001
3MH/creatinine ^a^ (µmol/mmol)	19 ± 3	40 ± 3 *	35 ± 4 *	32 ± 2 *	<0.003
**EDL**					
Weight (g)	0.23 ± 0.01	0.25 ± 0.01	0.26 ± 0.02	0.24 ± 0.01	0.40
Protein content (mg/g)	158 ± 8	164 ± 6	152 ± 6	179 ± 8	0.07
Protein content (mg)	36 ± 2	35 ± 2	35 ± 2	40 ± 2	0.28
**Soleus**					
Weight (g)	0.23 ± 0.02	0.23 ± 0.01	0.25 ± 0.02	0.23 ± 0.01	0.69
Protein content (mg/g)	165 ± 24	151 ± 9	170 ± 15	180 ± 38	0.80
Protein content (mg)	31 ± 3	34 ± 2	42 ± 4	39 ± 4	0.07
**Tibialis**					
Weight (g)	0.94 ± 0.03	0.99 ± 0.03	1.07 ± 0.06 *	1.11 ± 0.05 *	0.058
Protein content (mg/g)	177 ± 5	162 ± 7	179 ± 7	187 ± 9	0.13
Protein content (mg)	167 ± 7	164 ± 7	190 ± 14	199 ± 19	0.13
**Liver**					
Weight (g)	13.1 ± 0.8	17.3 ± 0.9 *	16.5 ± 0.8 *	16.0 ± 0.8 *	0.01
Protein content (mg/g)	140 ± 20	160 ± 10	160 ± 10	190 ± 10	0.11
**Plasma amino acids**					
ΣBCAA (µmol/l)	526 ± 47	587 ± 32	599 ± 16	551 ± 37	0.43
ΣEAA (µmol/l)	874 ± 53	940 ± 46	978 ± 29	910 ± 44	0.39
ΣNEAA (µmol/l)	2968 ± 75	2872 ± 155	2781 ± 147	2757 ± 99	0.66
ΣAA (µmol/l)	3842 ± 101	3811 ± 195	3759 ± 161	3667 ± 104	0.86

Malnutrition was induced in old rats by a dietary restriction to 50% of their spontaneous intakes for 6 weeks. Thereafter, eight rats (*n* = 8, DEN group) were sacrificed immediately, while the animals of the other groups received, for 4 days, their standard diet at 100% of their spontaneous food intake alone (*n* = 9, REN group) or with NCP, 1 mg/kg/d (*n* = 9, NCP1 group) or 10 mg/kg/d (*n* = 8, NCP10 group). All the parameters (means ± SEMs) were measured at the end of the 4-day feeding period in the fasted state or ^a^ from urine collected on the last 4th day. 3MH: 3-methylhistidine; EDL: extensor digitaris longus; ΣBCAA: sum of branched-chain amino acids; ΣEAA: sum of essential amino acids; ΣNEAA: sum of nonessential amino acids; ΣAA: total plasma amino acid level. One-way ANOVA and post-hoc Tukey–Kramer; statistical significance: *p* < 0.05. * *p* < 0.05 versus DEN.

**Table 6 nutrients-11-02637-t006:** The effect of NCP on muscle and liver AAs during the refeeding of malnourished old rats.

	DEN	REN	NCP1	NCP10	*p* Values
**EDL**					
Gln	4.90 ± 0.20	4.70 ± 0.10	5.50 ± 0.20 ^$^	4.70 ± 0.20 ^#^	<0.05
Val	0.17 ± 0.01	0.22 ± 0.01 *	0.22 ± 0.01 *	0.20 ± 0.01	<0.05
BCAA	0.43 ± 0.03	0.52 ± 0.02	0.52 ± 0.02	0.49 ± 0.03	0.099
Phe	0.06 ± 0.00	0.07 ± 0.00	0.07 ± 0.00 ^$^	0.06 ± 0.00 ^#^	<0.05
Cys	0.040 ± 0.000	0.036 ± 0.002 *	0.040 ± 0.000 ^$^	0.039 ± 0.001 ^$^	<0.02
Ser	1.16 ± 0.06	0.95 ± 0.03	1.12 ± 0.05	1.07 ± 0.08	0.067
Pro	0.22 ± 0.01	0.30 ± 0.02 *	0.30 ± 0.02 *	0.29 ± 0.02 *	<0.05
Gly	3.12 ± 0.17	2.49 ± 0.11 *	2.45 ± 0.16 *	2.44 ± 0.22 *	<0.05
**Soleus**					
Met	0.03 ± 0.00	0.04 ± 0.00	0.04 ± 0.01	0.04 ± 0.00	0.077
Val	0.16 ± 0.01	0.22 ± 0.01	0.21 ± 0.01	0.20 ± 0.01	0.071
Ile	0.08 ± 0.00	0.11 ± 0.06	0.10 ± 0.01	0.10 ± 0.01	0.087
Leu	0.17 ± 0.01	0.22 ± 0.01 *	0.21 ± 0.01 *	0.21 ± 0.01 *	<0.05
BCAA	0.42 ± 0.02	0.55 ± 0.03 *	0.52 ± 0.03 *	0.51 ± 0.03 *	<0.05
Glu	4.21 ± 0.52	3.14 ± 0.49	2.22 ± 0.32 *	3.29 ± 0.45	<0.05
Ala	1.94 ± 0.26	2.48 ± 0.11	2.35 ± 0.08	2.55 ± 0.20	0.08
Pro	0.23 ± 0.02	0.31 ± 0.02 *	0.28 ± 0.02	0.30 ± 0.01 *	<0.05
Gly	188 ± 30	177 ± 22 *	197 ± 34 *	174 ± 24 *	<0.05
**Tibialis**					
Val	0.17 ± 0.01	0.23 ± 0.02 *	0.21 ± 0.01	0.19 ± 0.02 ^$^	<0.05
Leu	0.18 ± 0.01	0.22 ± 0.02	0.20 ± 0.01	0.17 ± 0.02	0.098
BCAA	0.44 ± 0.03	0.57 ± 0.04 *	0.51 ± 0.02	0.45 ± 0.04 ^$^	<0.05
Tyr	0.09 ± 0.01	0.11 ± 0.01	0.09 ± 0.05	0.08 ± 0.01	0.084
Glu	0.84 ± 0.06	1.19 ± 0.08 *	1.14 ± 0.07 *	0.92 ± 0.06 ^$,#^	<0.05
Pro	0.24 ± 0.01	0.34 ± 0.02 *	0.31 ± 0.02 *	0.27 ± 0.02 ^$^	<0.05
Gly	3.52 ± 0.20	2.90 ± 0.20	2.70 ± 0.30 *	2.50 ± 0.20 *	<0.05
**Liver**					
Orn	0.45 ± 0.04	0.31 ± 0.03 *	0.34 ± 0.03	0.32 ± 0.02 *	<0.05
Gln	4.90 ± 0.20	4.80 ± 0.30	5.10 ± 0.20 ^$^	4.90 ± 0.30 ^#^	<0.05
Val	0.31 ± 0.04	0.29 ± 0.03 *	0.32 ± 0.01 *	0.31 ± 0.02	<0.05
Cys	0.102 ± 0.001	0.098 ± 0.002 *	0.100 ± 0.001 ^$^	0.099 ± 0.001 ^$^	<0.04
Ser	0.60 ± 0.10	0.50 ± 0.10	0.60 ± 0.10	0.60 ± 0.10	0.067
Ala	2.10 ± 0.20	2.80 ± 0.20 *	3.20 ± 0.30 *	3.20 ± 0.20 *	<0.05
His	0.60 ± 0.03	0.67 ± 0.05	0.72 ± 0.03	0.68 ± 0.03	0.092
Gly	2.90 ± 0.20	2.10 ± 0.20 *	2.00 ± 0.20 *	2.00 ± 0.20 *	<0.05

Malnutrition was induced in old rats by a dietary restriction to 50% of their spontaneous intakes for 6 weeks. Thereafter, eight rats (*n* = 8, DEN group) were sacrificed immediately, while the animals of the other groups received, for 4 days, their standard diet at 100% of their spontaneous food intake alone (*n* = 9, REN group) or with NCP, 1 mg/kg/d (*n* = 9, NCP1 group) or 10 mg/kg/d (*n* = 8, NCP10 groups). EDL: extensor digitaris longus. Results (means ± SEMs) are expressed as µmol/g tissue wet weight. One-way ANOVA and post-hoc Tukey-Kramer; statistical significance: *p* < 0.05. * *p* < 0.05 versus DEN, ^$^
*p* < 0.05 versus REN, ^#^
*p* < 0.05 versus NCP1.

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
