# Peer review of "Is N-Carbamoyl Putrescine, the Decarboxylation Derivative of Citrulline, a Regulator of Muscle Protein Metabolism in Rats?"

_nutrients, 2019, doi:10.3390/nu11112637_

Round 1
Reviewer 1 Report
This paper discusses that the effects of N-carbamoyl putrescine (NCP) intake on muscle protein metabolism in young adult and malnourished old rats. This paper also discusses that the toxicity of NCP intake. The main contribution of the paper is to indicate that NCP intake was associated with improving muscle protein metabolism through repression of muscle protein breakdown. However, a major revision of manuscript is needed before it can be accepted for publication.
Major comments:
The statistical analysis in this paper is partly unsuitable. Fisher’s PLSD test is unsuitable for comparison of more than 4 groups (Study 2). Comparison of protein content of muscle tissue is unsuitable for discussion of muscle protein metabolism. Protein content of muscle tissue is strongly affected by water content. Total protein amount of muscle tissue (protein content × muscle weight) is suitable for discussion of muscle protein metabolism. In study 1, NCP intake seems to affect the metabolism of slow twitch muscle, while that seems to be affect fast twitch muscle in study 2. It must be discussed about the discrepancy. Line 295: I think it is overestimation that the effects of NCP intake on muscle protein synthesis because no significant differences versus REN group were detected on muscle protein synthesis and mTOR signaling activity. If the authors discuss about insulin resistance, the data of plasma insulin in the fasting state is also needed. In study 1, the results of EDL should be described in Table 2.
Minor comments:
The sex of rats is not described in this paper. The age of rats in study 2 described in Materials and Methods is inconsistent with the description in Abstract. Table numbers in the manuscript are inconsistent with that in Table title. Table 1: Not NCP10 but NCP50. Table 2, 3: The meaning of asterisk should be described. Table 3: Unit should be described. Table 5: SEM is unclear. The unit of Nitrogen balance may be error. The meaning of footnotes ‘a’ should be described. Line 309: The plasma glutamine levels in NCP50 decreased compared with control group (Line 186).
Author Response
We thank the reviewers and the editor for their comments and have modified the manuscript accordingly.
Reviewer 1:
Major comments:
The statistical analysis in this paper is partly unsuitable. Fisher’s PLSD test is unsuitable for comparison of more than 4 groups (Study 2).We agree with this reviewer on the limitations of using the PLSD test. However, in the first study, we study only 3 groups and in the second only 4 groups (and not more). The PLSD test is actually recommended for comparison between 3 groups and, although relatively liberal, it is still applicable for 4 groups. In addition, it allows a homogeneity of the statistical analysis for the whole work.
Comparison of protein content of muscle tissue is unsuitable for discussion of muscle protein metabolism. Protein content of muscle tissue is strongly affected by water content. Total protein amount of muscle tissue (protein content × muscle weight) is suitable for discussion of muscle protein metabolism.
We agree with this reviewer that for the study of protein content in tissues, variations in hydration can make the data unsuitable for comparison. In addition to the fact that hydration was not modified in our experiments, statistical differences were observed in a similar way regardless of the mode of expression of muscle proteins. This is now indicated in the manuscript (see lines 121-122).
In study 1, NCP intake seems to affect the metabolism of slow twitch muscle, while that seems to be affect fast twitch muscle in study 2. It must be discussed about the discrepancy
Indeed, the choice of the different muscles is based on their different composition in terms of fibre types, but it is only a relative notion since none of these muscles contains exclusively a single type of fibre. In addition, fast-twitch muscles (EDL, tibialis) are more influenced by dietary restrictions than slow twitch muscles (soleus). Moreover, among the various factors that affect muscle homeostasis with age, it is known that the respective percentage of different types of fibre is affected by aging, while older rats have a lack of adaptation to long-term dietary restrictions. It would therefore be interesting to study which type of fibre is specifically affected. This discrepancy is now discussed (see lines 372-378).
Line 295: I think it is overestimation that the effects of NCP intake on muscle protein synthesis because no significant differences versus REN group were detected on muscle protein synthesis and mTOR signaling activity.
We agree and the manuscript has been modified accordingly (lines 312-312).
If the authors discuss about insulin resistance, the data of plasma insulin in the fasting state is also needed
We agree that it would probably have been preferable to have plasma insulin if blood glucose levels had been determined at any time. But it is the fasting blood glucose that we have measured, and which is abnormally high, which is clear evidence of glucose intolerance and therefore insulin resistance. This is now indicated in the manuscript (see lines 388-390).
In study 1, the results of EDL should be described in Table 2.
Sorry for that, but unfortunately (probably due to our standard laboratory procedures specifically for young adults or elderly rats), EDLs were not collected in study 1. This difference between studies 1 and 2 is now mentioned in the methods section (line 112).
Minor comments:
The sex of rats is not described in this paper.The sex of the rats is now mentioned in the abstract and the methods section (lines 18, 20, 69).
The age of rats in study 2 described in Materials and Methods is inconsistent with the description in Abstract.
Sorry for that: young adult rats were 8-week-old and aged rats 18-month old; this has been fixed (lines 20 and 69.
Table numbers in the manuscript are inconsistent with that in Table title.
We apologize for these mistakes. Table numbers in the text and in titles are now consistent.
Table 1: Not NCP10 but NCP50.
This has been corrected.
Table 2, 3: The meaning of asterisk should be described.
The meaning of the asterisk is now clearly described.
Table 3: Unit should be described.
In fact, the units were described in the table legend: μmol/l for plasma or nmol/g for tissue. They are now indicated in the table itself.
Table 5: SEM is unclear. The unit of Nitrogen balance may be error. The meaning of footnotes ‘a’ should be described.
As mentioned in the table legend, all data are presented as mean ± SEM.
The Nitrogen balance is indeed expressed in mg/24h. This has been corrected.
The meaning of footnotes ‘a’ is now described.
Line 309: The plasma glutamine levels in NCP50 decreased compared with control group (Line 186).
This has been corrected (line 327).
Reviewer 2 Report
With great interest I read the manuscript by Jegatheesan et al investigating the influence of NCP as a regulator of muscle protein synthesis.
General comment:
A minor English spell and grammar check should be performed (typo's, differences in past vs. present tense, certain words (ad libitum) etc should be in italics) .
Major comments:
1. A schematic representation of the investigated metabolism should be included in the introduction to clarify this for the readers.
2. The aim of the study is mentioned in the introduction, however, a clear hypothesis is missing.
3. In the materials and methods section a lot references to previously presented methods are found. However, there were adaptations were made to a protocol (line 147), these are mentioned. Please include this.
4. Has food intake been measured during the experiment? If so, was it comparable between the groups?
4. The result section is very unclear. For instance, in line 160 table 2 is referred. This presumably should be table 1. Wrongful referencing to the tables continues throughout the entire result section making it unreadable. Therefore, conclusions and points of discussion is section 4 can also not be assessed adequately.
Please re-write the entire result section!
5. The lay-out of tables from the second study is not clear. If this is not due to the formatting lay-out of the journal, please improve.
6. Please include representative western blot images.
Minor comments:
-Abstract: different abbreviations are not explained (EDL, S6K1 etc). Please introduce them on first use.
-Line 22: "NCP had no influence on weight gain", in which experiment?
-Line 100: Why are jejunum and ileum tissues only collected in young animals?
-Line 133: What is the species origin of the different antibodies?
-Subheading 3.2 should be placed one paragraph earlier.
Author Response
We thank the reviewers and the editor for their comments and have modified the manuscript accordingly.
Reviewer 2:
General comment:
A minor English spell and grammar check should be performed (typo's, differences in past vs. present tense, certain words (ad libitum) etc should be in italics).
The use of language has been checked
Major comments:
A schematic representation of the investigated metabolism should be included in the introduction to clarify this for the readers.A new figure has been added with the AA amine derivatives metabolism
The aim of the study is mentioned in the introduction, however, a clear hypothesis is missing
As the amines derived from ornithine and arginine that participate in the metabolic properties of these amino acids, NCP could participate in the anabolic effects of citrulline. The hypothesis of this study is now presented (line 56-58).
In the materials and methods section a lot references to previously presented methods are found. However, there were adaptations were made to a protocol (line 147), these are mentioned. Please include this.
The technique used is described in detail and its performance is evaluated in ref. 10. The sentence has been slightly modified to better present this point (lines 159-160).
Has food intake been measured during the experiment? If so, was it comparable between the groups?
As already mentioned in the manuscript, the animals were fed 100% of their daily intakes measured during the acclimatization period. Complete food consumption was verified in all experiments. This is now mentioned more clearly (lines 73-75 and line 96).
The result section is very unclear. For instance, in line 160 table 2 is referred. This presumably should be table 1. Wrongful referencing to the tables continues throughout the entire result section making it unreadable. Therefore, conclusions and points of discussion is section 4 can also not be assessed adequately. Please re-write the entire result section!
We apologize for these errors related to the deletion of a table at the last minute before the submission of the manuscript. We hope that the results section is now clear; the numbers of the tables in the text are now consistent with those of the table titles.
The lay-out of tables from the second study is not clear. If this is not due to the formatting lay-out of the journal, please improve.
This was related to the use of tabulations in tables. These have been removed and the tables are now easier to read.
Please include representative western blot images.
This has been done
Minor comments:
- Abstract: different abbreviations are not explained (EDL, S6K1 etc). Please introduce them on first use.
Most abbreviations have been removed from the abstract and the remaining is explained.
- Line 22: "NCP had no influence on weight gain", in which experiment?
NCP has no effect on weight gain in either of the 2 studies; this has been clarified (line 22).
- Line 100: Why are jejunum and ileum tissues only collected in young animals?
Jejunum and ileum tissues were only collected in young animals because the aim of our first study was to evaluate NCP tolerance and its effect on polyamine metabolism whereas in the second study we focused on the effect on muscle metabolism. The respective objectives of the 2 parts of our work are explained in the abstract (see lines 16-17) and in the introduction (lines 58-61).
-Line 133: What is the species origin of the different antibodies?
The species origin of the different antibodies is now indicated (lines 142-144).
-Subheading 3.2 should be placed one paragraph earlier
All headings and subheadings have been checked and corrected when necessary.
Round 2
Reviewer 1 Report
The authors revised the manuscript in response to reviewer’s comments, but there are still some points to be improved.
Major comments:
Fisher's LSD test should be restricted to 3 groups. I highly recommend that the authors use Tukey-Kramer test for post hoc analysis in Study 2. I highly recommend that total protein amounts of each muscle tissue (protein content (mg/g) × muscle weight (g)) are added in Table 2 & 5. The data will be useful for discussion of the effect of NCP on muscle protein metabolism. High levels of fasting blood glucose are caused by not only insulin resistance but also reduction of insulin secretion, so the expression ‘glucose intolerance’ is suitable in this paper but ‘insulin resistance’ is not.Author Response
We thank the reviewer for his comments and have modified the manuscript accordingly.
Fisher's LSD test should be restricted to 3 groups. I highly recommend that the authors use Tukey-Kramer test for post hoc analysis in Study 2.
This has been done
I highly recommend that total protein amounts of each muscle tissue (protein content (mg/g) × muscle weight (g)) are added in Table 2 & 5. The data will be useful for discussion of the effect of NCP on muscle protein metabolism.
This has been done
High levels of fasting blood glucose are caused by not only insulin resistance but also reduction of insulin secretion, so the expression ‘glucose intolerance’ is suitable in this paper but ‘insulin resistance’ is not.
In fact, we were referring to the catch-up fat situation that is clearly associated with insulin resistance. However, we agree that a possible decrease in insulin secretion greater than that already induced by overnight fasting cannot be ruled out. The manuscript has been modified accordingly (see abstract and end of the discussion/conclusion).
Round 3
Reviewer 1 Report
The authors revised the manuscript in response to my comments.